# Multimodal tactile sensing fused with vision for dexterous robotic housekeeping

Qian Mao [1,2], Zijian Liao [1,2], Jinfeng Yuan [1] & Rong Zhu [1] ✉

As robots are increasingly participating in our daily lives, the quests to mimic human abilities have driven the advancements of robotic multimodal senses. However, current perceptual technologies still unsatisfied robotic needs for home tasks/environments, particularly facing great challenges in multisensory integration and fusion, rapid response capability, and highly sensitive perception. Here, we report a flexible tactile sensor utilizing thin-film thermistors to implement multimodal perceptions of pressure, temperature, matter thermal property, texture, and slippage. Notably, the tactile sensor is endowed with an ultrasensitive (0.05 mm/s) and ultrafast (4 ms) slip sensing that is indispensable for dexterous and reliable grasping control to avoid crushing fragile objects or dropping slippery objects. We further propose and develop a robotic tactile-visual fusion architecture that seamlessly encompasses multimodal sensations from the bottom level to robotic decision-making at the top level. A series of intelligent grasping strategies with rapid slip feedback control and a tactile-visual fusion recognition strategy ensure dexterous robotic grasping and accurate recognition of daily objects, handling various challenging tasks, for instance grabbing a paper cup containing liquid. Furthermore, we showcase a robotic desktop-cleaning task, the robot autonomously accomplishes multi-item sorting and cleaning desktop, demonstrating its promising potential for smart housekeeping.

In the field of robotics, mimicking human sensory capabilities has been a persistent pursuit, driven by the desire to endow machines with a deeper understanding of the surrounding world. In recent years, tactile sensing and the fusion of tactile and visual senses (referred to as tactile-visual fusion sensing) have emerged as groundbreaking approaches in this endeavor. These technologies hold the potential to revolutionize the way robots perceive their environment and interact with it[1,2]. Incorporating tactile sensing into robotics signifies a transformative shift in machine capabilities, granting them the ability to perceive and cognize their surroundings through touch, much akin to humans[3–6]. Traditional robots have heavily relied on visual perception, while seldom considering the tactile senses that are necessary for interaction in complex and dynamic environments[7–10]. Leveraging principles such as piezoelectric[11–14], piezoresistive[15–19], triboelectric[3,20–22], capacitive[5,23–25],

and thermosensitive[26–28], tactile sensors fuse various tactile attributes such as pressure, temperature, texture, and material properties, providing robots with rich senses and enabling them to interact with the environment more dexterously. In manufacturing, robots equipped with tactile sensors would execute fine assembly tasks with higher precision and adaptability[29]. In healthcare, tactile feedback potentially aids minimally invasive surgeries to surgeons, enhancing the precision of surgical instrument manipulation[30]. However, recent tactile sensing technology still unsatisfied most demands for assistive living service robots in terms of multisensory integration and fusion, fast response capability, and highly sensitive perceptions.

What's more, for intricate robotic tasks, sole tactile perception faces challenges such as a lack of visual guidance and an inability to locate objects in the environment[1,31]. Besides, sole visual perception

[1]State Key Laboratory of Precision Measurement Technology and Instruments, Department of Precision Instrument, Tsinghua University, Beijing, China. [2]These authors contributed equally: Qian Mao, Zijian Liao. ✉e-mail: zr_gloria@mail.tsinghua.edu.cn

encounters issues like ambient light interference, occlusion, and confusion with similar-shaped objects[1,2,32–36]. For example, the accuracy of visual recognition is poor for these visually similar objects of plastic bags, napkins, and wrapping paper which are common in our daily lives. Therefore, integrated utilization of tactile and visual senses through tactile-visual fusion, a holistic approach to understanding objects and situations is needed to mirror the human ability to integrate sensory inputs for enhancing comprehension[34,36,37]. Machine learning algorithms play a pivotal role in facilitating the integration of these diverse sensory inputs within tactile-visual fusion[6,8,35,38]. For example, deep learning architectures such as convolutional neural networks (CNNs) and recurrent neural networks (RNNs) exhibit exceptional capabilities in capturing intricate correlations between multimodal sensing data and objects to be recognized[1,8,39]. Advancements in tactile-visual fusion algorithms enhance object recognition, refine grasping strategies, and enable more precise control over robotic manipulation devices. These tactile-visual fusion technologies empower robots with multimodal perception abilities, allowing them to perceive their surroundings in multiple dimensions similar to humans, thereby accomplishing more complex and challenging tasks[1,8,9,34,39,40].

However, there are still many factors influencing the ability of robots to perform intricate tasks in real scenarios, and thus hard to replace human labor. Firstly, there's the challenge of enabling robots with human-like sensory capabilities[41,42]. Take the example of pouring hot water from a cup – in this process, a robot requires visual perception for cup localization, and it needs tactile information such as pressure, temperature, and slippage to execute the pouring action[32,40].

Just like humans, for robots to interact effectively with their environment, multimodal sensing abilities like vision, pressure, temperature, thermal attributes, textures, and slipping are all indispensable[10,35,36]. Furthermore, fast and sensitive perceptive capabilities are also imperative. For instance, lacking a great ability for fast and sensitive slip detection might result in a robot dropping an object while trying to grasp it, and causing losses or endangering individuals. Lastly, a universally applicable and generalizable design for fusing diverse sensory information remains elusive[1,2,8,10,31,33,35,36,40,43]. Multi-modal sensory information often originates from a variety of sensors, each with distinct data types, frequencies, and ranges. The challenge persists in terms of how to integrate these sensing inputs at various levels – data layer, feature layer, decision-making layer, etc.[1]

In this paper, we propose a robotic flexible tactile sensor based on thermosensation, enabling simultaneous multimodal perceptions of contact pressure, temperature, thermal conductivity, texture, and slippage (Fig. 1). It is worth mentioning that the tactile sensor has an ultrasensitive (0.05 mm/s) and ultrafast response (4 ms) to slip detection which is indispensable for fine grasping control. Furthermore, we propose a tactile-visual fusion robot architecture that seamlessly encompasses multimodal senses at the bottom level to robotic decision-making at the top level, which autonomously accomplishes object location, stable grasping, and object recognition. The proposed tactile-visual fusion grasping strategy is utilized for grasping various objects, where vision assists in determining object position and pose, and a slipping-based tactile feedback control ensures dexterous robotic grasping. It is also worth mentioning that the robot is able to stably grasp the cup with minimal strength by

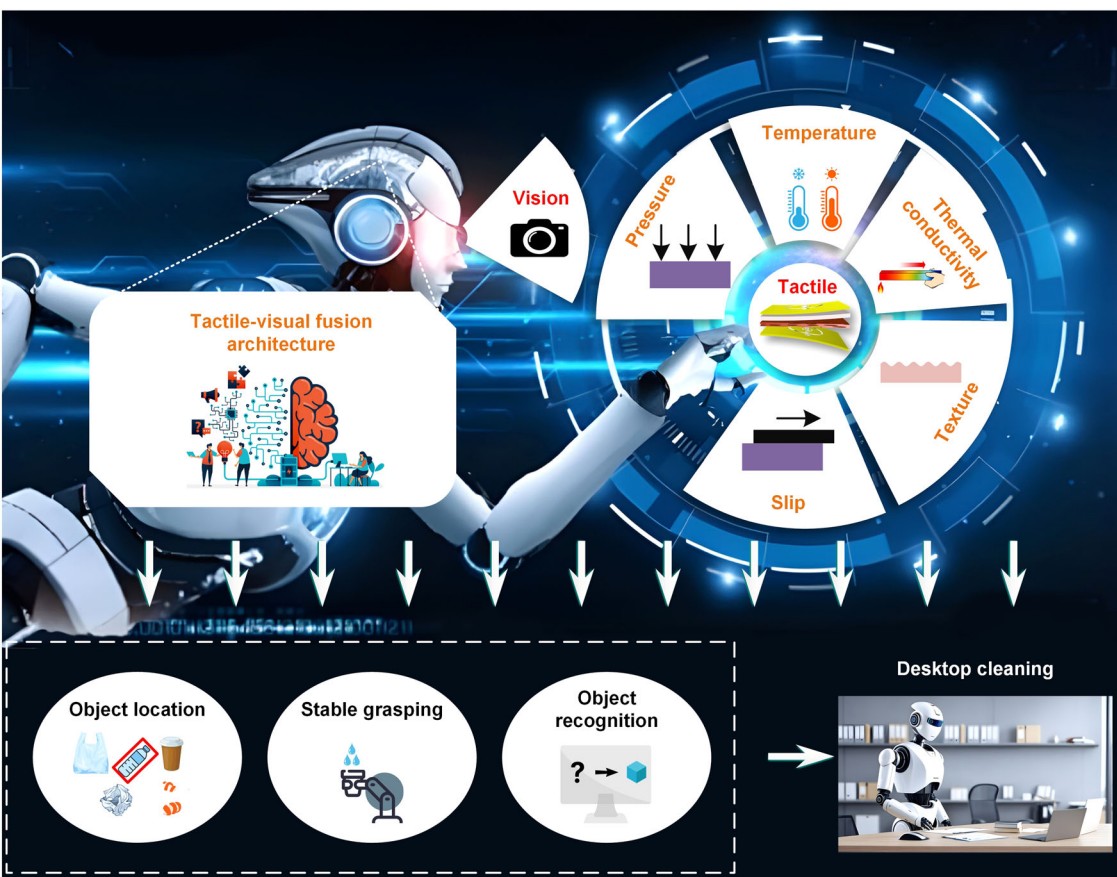

**Fig. 1 | A tactile-visual fusion robot enables complex tasks.** The robot has multimodal perception capabilities, including vision and touch (perceptions of contact pressure, temperature, thermal conductivity and texture of objects, and slip). Enhanced by these perceptual capabilities and bolstered by the proposed tactile-visual fusion strategy, the robot can accomplish a series of tasks such as object location, stable grasping, object recognition, and sorting.

tactile sensing to ensure neither crushing the cup nor slipping off it while pouring water into it. In addition, we further propose a tactile-visual fusion recognition strategy using a tailor-designed cascade classifier to achieve accurate recognition of various objects, which is significantly superior to either of tactile or visual recognition method. Moreover, we apply the proposed tactile-visual fusion method to a desktop-cleaning task, the robot automatically accomplishes a series of actions such as object location, stable object grasping, and accurate object recognition to achieve multi-item sorting and cleaning on the desktop (Movie S1). It validates the feasibility and superiority of the proposed multimodal tactile sensor as well as tactile-visual fusion architecture, demonstrating the promising potential of smart robots for housekeeping applications.

## Results

### Working principles of the multimodal flexible tactile sensor

Our proposed flexible tactile sensor employs unified simple thermometry to perceive multimodal attributes, including pressure, temperature, thermal conductivity, and texture of objects, as well as slip detection. Each tactile sensor consists of a top sensing layer, a bottom sensing layer, a PDMS layer, and a porous material in the middle (Fig. 2a). The top and bottom sensing layers have the same sensing structure, which encompasses two concentric platinum (Pt) thermistors deposited on a flexible polyimide substrate. The inner thermistor has a low resistance (~50 Ω), which acts as a thermal element (denoted as hot film). While the outer thermistor has a higher resistance (~500 Ω, denoted as cold film), working as a local temperature sensor. The radii

of the hot film and cold film are 0.95 mm and 3.2 mm, respectively (Fig. 2a). The sensing principle is based on conductive heat transfer. The hot film on the sensing layer serves as both a Joule heater and its own temperature detector, which is sensitive to its surrounding heat conduction.

For the bottom sensing layer, contact pressure generates an elastic deformation of the middle porous material, altering the thermal conductivity of the porous material due to its piezo-thermic transduction[44] (Fig. 2b). Consequently, the bottom sensing layer responds to the contact pressure (the output signal is denoted as $U_p$). Figure 2c shows the response curve of the sensor under pressure loading and unloading. The sensor has a broad measuring range of 20 N and a low detection limit of 0.01 N, and exhibits a low hysteresis of 2.4%.

Regarding the top sensing layer, when the sensor comes into contact with an object, the thermal conductivity of the object influences the heat transfer of the hot film and thus be detected. Specifically, the hot film in the top sensing layer is electrically heated by the constant temperature difference (CTD) circuit shown in Fig. S1 to ensure its temperature is higher than the surroundings and generates a thermal field in the contacted object. Upon slippage, the hot film shifts to a cooler area on the object that it's in contact with, leading to a change in the heat transfer, which is detected immediately by the hot-film. In essence, slippage is identified by the alteration in the heat transfer of the hot film. After a slip event, the heat transfer reverts to its pre-slippage state, as the sensor remains in contact with the heated area. Therefore, the slipping of the object is detected with the output

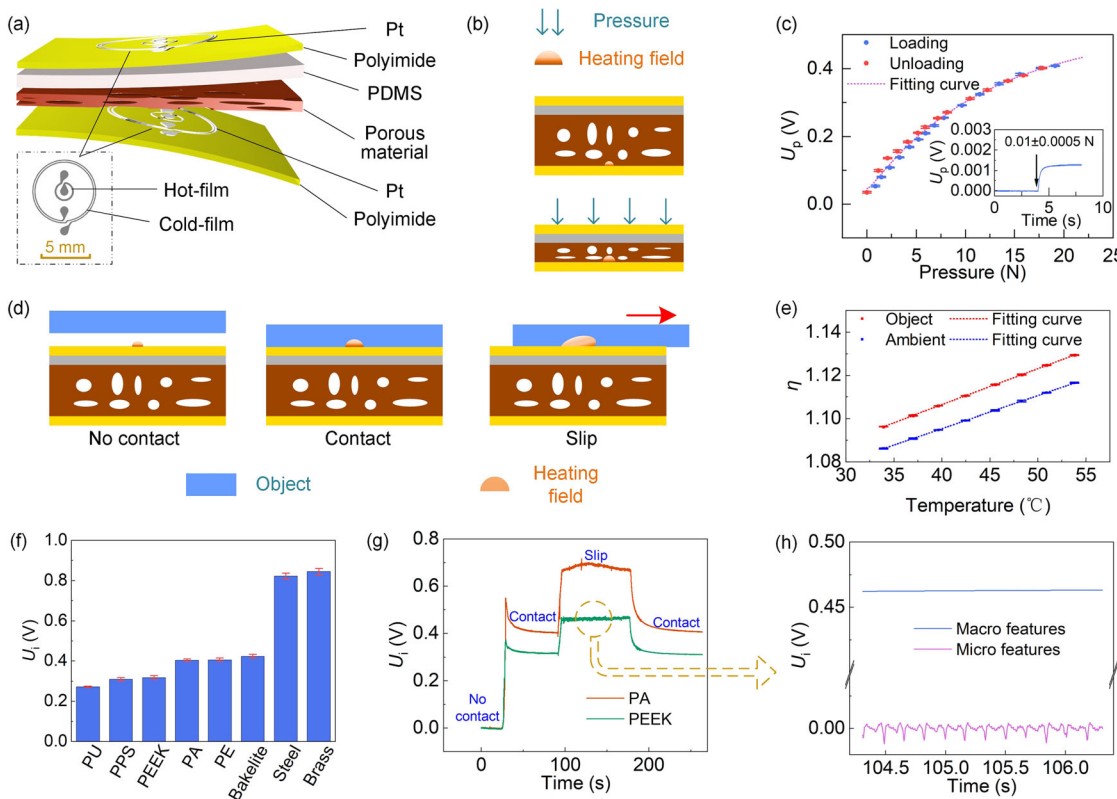

**Fig. 2 | The structure, working principle, and functions of the multimodal tactile sensor. a** The structure of the tactile sensor is composed of a top sensing layer, a bottom sensing layer, PDMS, and a porous material in the middle. **b** Working principle of the bottom sensing layer. **c** The pressure response $U_p$ of the bottom sensing layer under loading and unloading. Inset shows the low detection limit. **d** Working principle of the top sensing layer. The heat transfer of the top sensing layer under the states of no-contact, contact, and slip. **e** The output signals of the tactile sensor responding to ambient temperature and object temperature,

respectively. **f** The responses of the top sensing layer when touching on materials with different thermal conductivities. **g** The responses of the top sensing layer when contacting with and slipping on PA and PEEK, respectively. **h** Slip and texture are detected from the macro and micro features of the signals of the top sensing layer. The macro and micro features are extracted from the signals of the top sensing signal by filtering during the slipping process. Error bars shown in **c**, **e**, and **f** are the standard deviations of five repeated measurements of one tactile sensor.

signal of the top sensing layer (denoted as $U_i$) (Fig. 2d). In addition, as mentioned before, the cold-film thermistors in the bottom sensing layer and the top sensing layer perceive the ambient temperature and object temperature respectively, as shown in Fig. 2e:

$$\eta_{ambient} = \frac{U_{pc}}{U_p - U_{pc}} \qquad (1)$$

$$\eta_{object} = \frac{U_{ic}}{U_i - U_{ic}} \qquad (2)$$

where $U_{pc}$ and $U_{ic}$ refer to the cold-film voltages of the top sensing layer and bottom sensing layer, respectively). Utilizing the CTD circuit, the tactile sensor is immune from environmental temperature variation[45] (details in Supplementary text and Figs. S1 and S2).

Figure 2f shows the change in the response of the top sensing layer when it touches objects with different thermal conductivities. The thermal conductivities of the above materials are provided in Table S1. It is seen that as the thermal conductivity of the object increases, the output signal of the sensor increases. Figure 2g shows the dynamic response of the sensor as it transitions from non-contact to contact, slipping, and back to contact with objects. Since the thermal conductivity of polyamide (PA) is larger, it is seen that the response during contact and sliding is greater than that of polyetheretherketone (PEEK) material. It is worth mentioning that the changes in contact pressure do not affect slip detection (Fig. S3). More importantly, since the sensor is sensitive to the surface topography of the object, the response during the slipping process is filtered (details in Methods) to extract its macro features ($U_{macro}$) and micro features ($U_{micro}$) (Fig. 2h). These features can be used to detect the slipping state and the texture of the object surface simultaneously as follows. Furthermore, the tactile sensor is tested in the reciprocating contact-separation cycles up to 1000 times (Figure S4), which indicates good stability and reliability of the sensor.

## Slip detection and texture recognition

Macro features in the response signals of the top sensing layer can be utilized to characterize the slipping state of the object on the sensor. We select different materials (Bakelite, PE, PA, PEEK, and PPS) for testing and record the sensor responses at slipping velocities of 0 (no sliding), 5, 10, 15, 20, 30, and 50 mm/s, respectively (Fig. 3a). The results indicate that as the thermal conductivity of the material increases and the slipping velocity becomes faster, the corresponding macro features ($U_{macro}$) increase. When the thermal conductivity of the object has been known from a prior steady contact state, $U_{macro}$ can be used to obtain the slipping velocity of the object during robotic grasping. Furthermore, we employ PPS material to determine the minimum detection threshold and response time for slip detection. As depicted in Fig. 3b, at a tiny slipping speed of 0.05 mm/s, the sensor still exhibits a distinct response. Besides, following the occurrence of slipping, the sensor's response surpasses the noise level of no-sliding conditions in just 4 ms (shown in Fig. 3c). These results collectively indicate that the proposed tactile sensor can be employed for ultra-sensitive and ultrafast slip detection across different materials.

Micro features in the response signals of the top sensing layer can be employed to detect the micro-surface morphology of objects, facilitating texture recognition. Figure 3d presents the micro-feature signals of the sensor as the PA material slips over the object's surface with a normal pressure of 2 N at various speeds (0.5, 1, 3, and 5 mm/s). With an increasing slipping speed, as the surface topography of the object remains constant, the peak number of signals within the same timeframe also rises. Performing a fast Fourier transform (FFT) analysis on these signals in the frequency domain yields the spectrum shown in Fig. 3e. It's evident from the graph that the frequencies corresponding to spectral peaks also increase with the rise in slipping speed. At a

slipping speed of 1 mm/s, the corresponding frequency of PA is 4.03 Hz, so the grating period on the surface of PA is approximately 248 μm, which is consistent with the micrograph in Figure S5. Visualizing the short-time Fourier transform (STFT) utilizing micro features offers time-dependent frequency information, depicting the magnitude intensity of the STFT across time (Fig. 3f). With an escalation in sliding speed, the frequency distribution gradually shifts towards the high-frequency region. Texture detection is also carried out for other different materials (details can be found in Figures S5 and S6).

Therefore, the thermal conductivity, slipping velocity, and surface texture of the object can be independently detected using the top hot-film sensing signal ($U_i$) from the contact to the slipping process. When the sensor comes into contact with the material, the response of the hot film in the top sensing layer changes significantly, and the stabilized voltage ($U_i$) can be used to determine the thermal conductivity of the material. When slipping occurs in robotic grasping process, the slipping velocity can be derived from the macro features ($U_{macro}$) extracted from the top hot-film sensing signal combined with the predetermined thermal conductivity. Besides, by performing FFT on the micro features ($U_{micro}$) extracted from the top hot-film sensing signal, the grating period of the material surface can be deduced to reflect the texture.

Using tactile sensors to measure the thermal property of the material and identify textures can also accurately differentiate among various fabrics. We select ten different fabrics (polyester spandex, polyester knitted stretch fabric, nylon, encrypted imitation silk, cotton canvas, denim, polar fleece, wool-polyester fabric, carton and linen fabric, and lycra, their photographs are shown in Fig. S7) to acquire the responses of the tactile sensor (Figs. 3g and S8) when it slides on the surfaces of these fabrics. In addition, the tactile response of the combination of fabrics (F1 and F3) is also shown in Fig. 3h, indicating that the sensor contacts two fabrics in sequence during a slipping process, and the corresponding macroscopic and microscopic characteristics of the sensor signal can be utilized to identify the fabrics in sequence. Subsequently, we extract macro features and micro features from the sensing data of the tactile sensor to conduct fabric recognition (details can be found in "Methods" section). Using this approach, we achieve accurate recognition of these 10 fabrics, with a total accuracy of 94.3% (Fig. 3i).

## Tactile-visual fusion architecture and stable grasping strategy

Although the proposed multi-modal tactile sensor bestows robots with rapid and intricate tactile perception, tactile sensing alone falls short in meeting the demands of robots in complex scenarios. Consequently, we combine tactile sensing with visual sensing and further introduce a hybrid tactile-visual fusion architecture. This architecture integrates tactile and visual sensing information at the data, feature, and decision levels, empowering robots with the capability to interact effectively with intricate environments. The specific architecture of the robot is shown in Fig. 4a. We divide the architecture into different levels, starting from the bottom with the signal level, followed by the perception level, decision level, and finally the system level. At the signal level, a binocular depth camera is used to capture visual signals, and the aforementioned multimodal tactile sensors are used to collect interface, slip, pressure, and temperature signals. For the perception level, the computer transforms the sensor signals into corresponding cognitions. Specifically, visual signals enable object recognition and object localization (details in "Methods" section, Figs. S9 and S10), while tactile signals allow for the contact perception of temperature, thermal conductivity, contact pressure, texture, and slipping state of objects. On the basis of multimodal perceptions, the robot accomplishes the corresponding decision and sends the task assignments to the actuators (robotic arm, robotic hand, and automated guided vehicle (AGV)). Actuators perform a series of action controls, such as the vehicle movement by the AGV, approaching an object, grasping

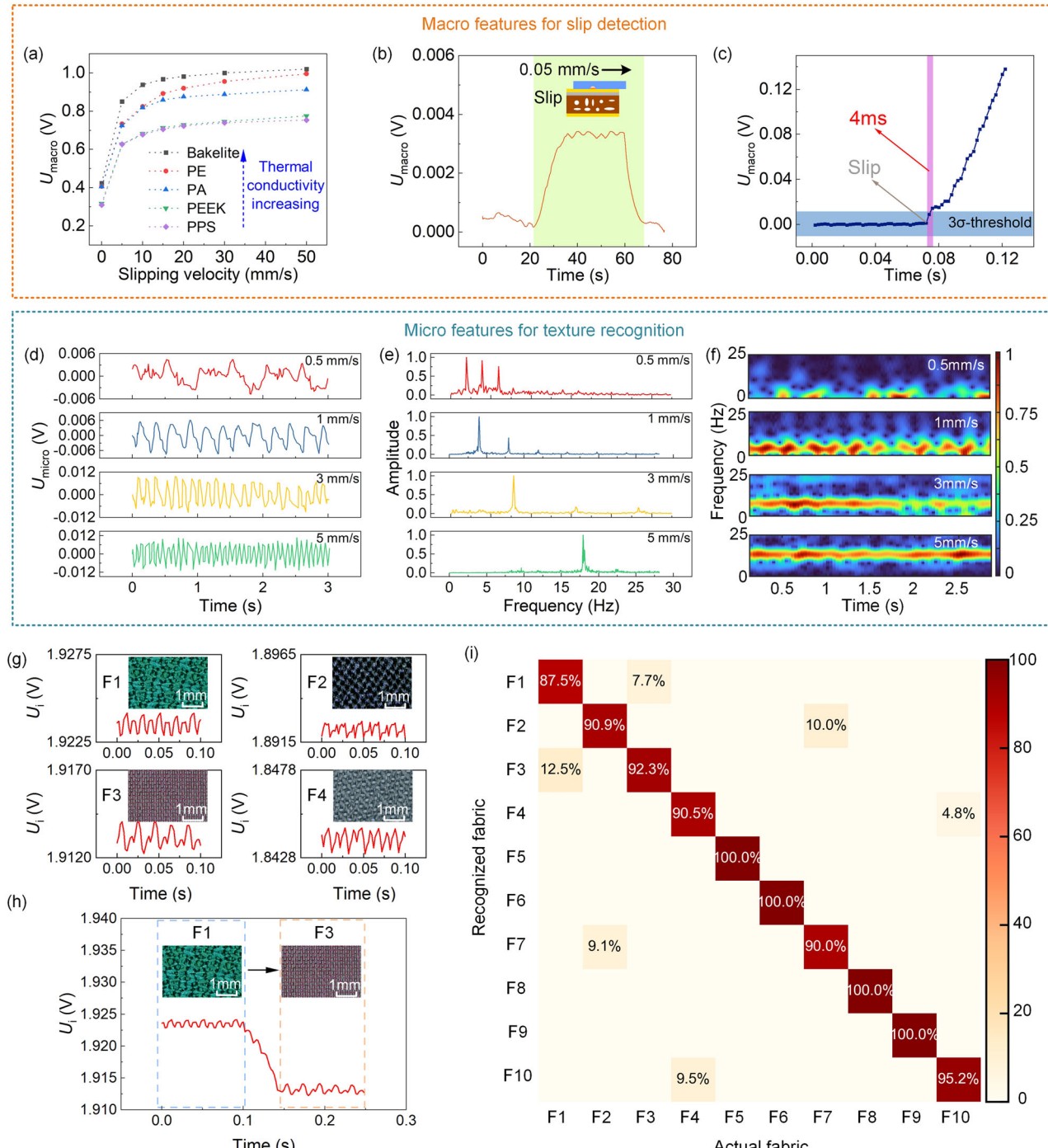

**Fig. 3 | Characteristics of slip detection and texture recognition. a** Macro features for different materials at different slipping velocities. **b** The tactile sensor has a low detection limit of 0.05 mm/s for slip detection. **c** The tactile sensor has a fast response time of 4 ms for slip detection. **d** Micro features at different slipping velocities. **e** FFT analyses of the micro features at different slipping velocities. **f** FFT analyses of the micro features at different slipping velocities. **g** Tactile responses and micrographs of four types of fabric including polyester spandex, polyester knitted stretch fabric, nylon, and encrypted imitation silk (F1–F4). The rest tactile responses are shown in Fig. S8. **h** The tactile response of a combination of F1 and F3. **i** The confusion matrix for fabric recognition.

and sorting of objects by the robotic arm and hand. By integrating all these levels, we have established a comprehensive tactile-visual fusion robot system architecture (system level). What's more, by incorporating additional sensors and actuators, it can empower robots with even more perception and execution capabilities, which enables the implementations of more intricate tasks.

In this architecture, we further propose a tactile-visual fusion grasping strategy to assist the robot in achieving a stable grip on a variety of objects. Due to the variety of shapes and sizes of objects, in order to achieve stable grasping, it is necessary to employ individualized grasping strategies according to the features of the objects. Common grasping strategies are mainly divided into model-based[46–48] and model-free methods[49,50]. Model-based methods generally formulate grasping strategies through pre-trained models, but they are associated with relatively high training costs[51]. The model-free method does not need to obtain the type information of the item and directly

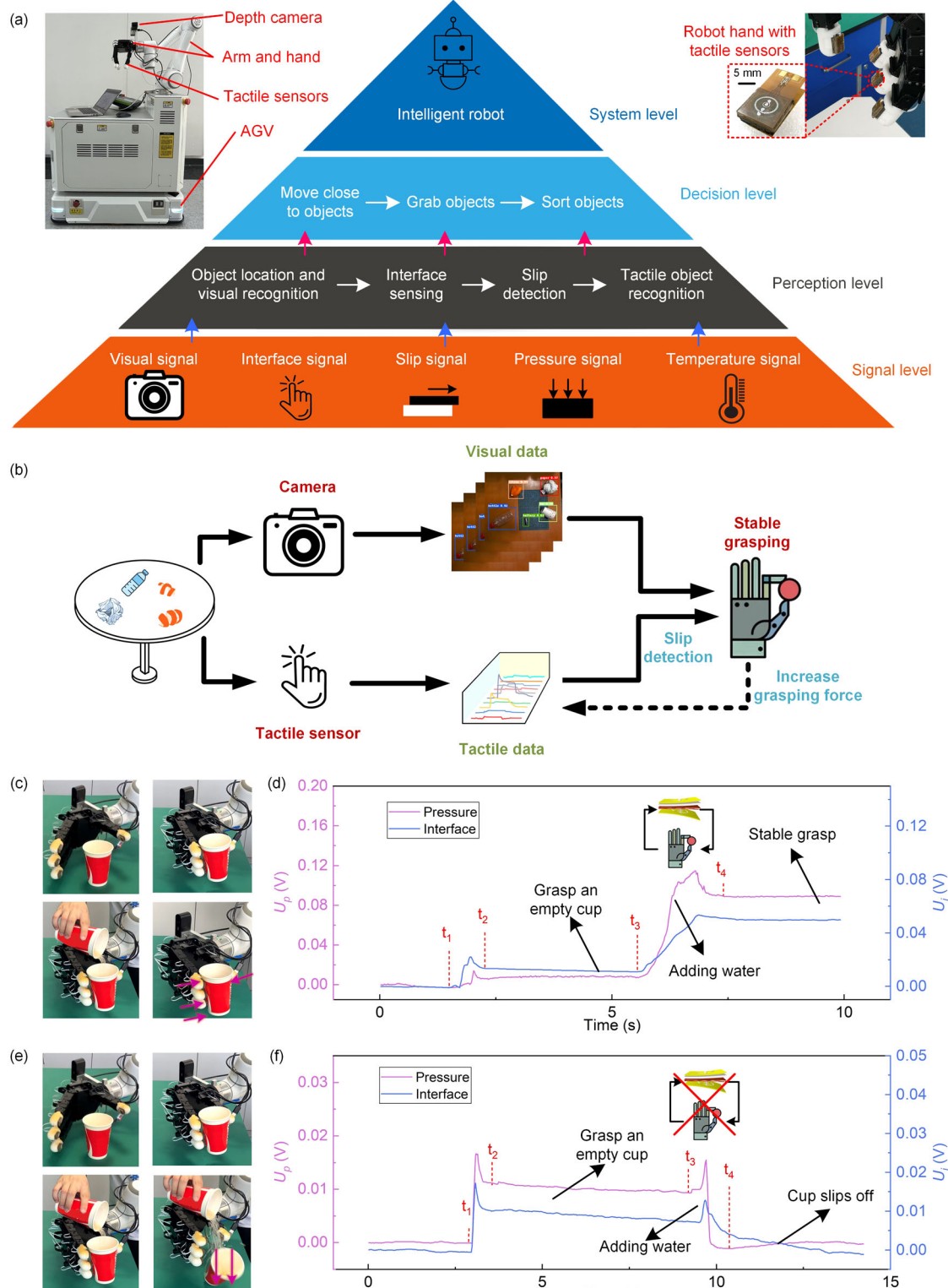

**Fig. 4 | The architecture of the tactile-visual fusion robot. a** A tactile-visual fusion robot architecture, including signal level, perception level, decision level, and system level. **b** A tactile-visual fusion grasping strategy for stable grasping: vision provides grasping position and pose, and tactile feedback control adjusts grasping force by slip detection in real-time. **c** Photographs of grasping a paper cup and adding water with tactile feedback. **d** The tactile signals during the grasping and adding water process. At the beginning, the robot hand stably grasps an empty cup.

Then water is added to the cup with the tactile feedback, and the robotic hand maintains a stable grip. **e** Photographs for grasping a paper cup and adding water without tactile feedback. **f** The tactile signal during the grasping and adding water process. At the beginning, the robot hand stably grasps the empty cup. Then water is added to the cup without tactile feedback, the cup finally slips off due to the increased weight.

determines the grasping strategy based on the observation results given by the camera, such as the commonly used five-dimensional grabbing method[50]. However, this method lacks detailed tactile information on the object, so it cannot perform precise operations. Here, we propose a tactile-visual fusion grasping strategy (Fig. 4b). First, the grasping position and pose of the robotic hand are determined according to the object's outline, size, and depth acquired by vision (Figure S11). When the robot is grasping an object, it executes a light grip at first and utilizes tactile sensing for real-time slip detection. When the slipping is detected, the robotic hand incrementally increases its grip force until reaches a stable hold. When no slipping is detected, the robotic hand maintains its current grip state. By employing this tactile feedback control, the grip strength applied by the robotic hand is minimized to the extent where slipping does not occur, which is particularly crucial when handling delicate or fragile objects.

To demonstrate that our grasping strategy can be applied to slippery or fragile objects, we utilize a robot hand equipped with the tactile sensors on its fingers to grasp a paper cup that is gradually pouring with water (Movie S2). As shown in Fig. 4c, d, in the initial state ($0$-$t_1$), the robot hand does not grasp any object, so the pressure signal and interface signal from the tactile sensor are both zero at this time. At the moment $t_1$-$t_2$, the hand starts to grasp an empty cup (weighs about 6.8 g), and the grip force increases gradually. After the hand completes the grasping action, the pressure and interface signals remain basically unchanged ($t_2$-$t_3$), which indicates a steady grip is maintained and no slip occurs. After that, water is poured into the cup at $t_3$, and a slip between the cup and the robot hand occurs due to the increased weight of the cup, and is quickly detected by the tactile sensors. The robot hand rapidly responds to increase the grip force under the real-time feedback control until no slip is detected, thus achieving a stable grip ($t_3$-$t_4$). Afterwards, the pressure signal and interface signal remain unchanged which indicates the pouring water stops at time $t_4$, and the grasp remains stable thereafter. At this time, the cup weighs ~100 g, which is about 15 times the original weight, while the paper cup with the water is stably held but undeformed. It is noted that a large grip force would crush the paper cup and thus spill the water. As a comparison, Fig. 4e and f show the results without the slip feedback control. When pouring water into the cup, the cup slips off as the robot hand is unaware of the slip and thus fails to adapt its grip force accordingly. This comparison demonstrates that the real-time slip feedback control can achieve stable grasping of objects while minimizing the grip force to avoid crushing fragile objects, which is indispensable for the delicate manipulation of robots. It is important that the slip detection should be ultrasensitive and ultrafast (in this work, the slipping detection reaches ultra-sensitivity of 0.05 mm/s and an ultra-fast response time of 4 ms) to ensure the success of above stable grasping.

### Tactile-visual fusion recognition strategy

In addition to stable object grasping, accurate object recognition is also an essential function of robots. For instance, when a robot assists in housekeeping tasks like serving drinks, it usually requires identification of a cup and determination of whether there is content inside, along with an approximate assessment of the content's composition for subsequent precise manipulation. In our daily life, humans usually identify objects through vision. However, robot vision has the limitation in recognizing objects in the home environment due to ambient light interference, occlusion, and confusion with similar-shaped objects as mentioned before. The objects of daily life have diverse materials, and many of them have similar shapes and similar colors. Vision alone struggles to distinguish between similarly shaped everyday objects, such as crumpled paper, plastic bags, and napkins. For objects that can't be recognized through vision alone, humans utilize tactile sensing to make precise judgments based on the object

attributes of temperature, pressure, thermal conductivity, texture, etc. Drawing inspiration from this concept, we propose a cascade tactile-visual fusion strategy for object recognition, which synthesizes multi-modal sensory information to accurately identify objects (Fig. 5a). Firstly, visual information is utilized with a YOLOv3 model to recognize objects based on their shape, size, color, etc., resulting in differentiable categories such as ball-shaped, bottle-shaped, cup-shaped, shapeless, etc. Subsequently, for visually similar objects within the same category, tactile sensing is employed for finer discrimination. And the shapeless objects could be categorized into types like plastic bag, wrapping paper, napkin, fabric, etc. through the utilization of thermal conductivity, pressure, and temperature information of the object by a shallow neural network (SNN). The fabric can be further subdivided into fleece, denim, nylon, etc. by using a bagging tree classifier according to its matter thermal conductivity and texture. As for cup-shaped objects, it's noteworthy that visual perception alone can't determine whether there are contents within an opaque cup. We can utilize the slip detection to determine if there is a weight inside the cup and further employ the attributes of thermal conductivity and temperature to make an assessment of the composition of the contents (details in Figs. S12 and S13). By following this approach, we effectively integrate multiple sensory inputs to achieve precise object identification. The recognition time using the tactile-visual fusion strategy is about 80 ms. Furthermore, as more sensory information is incorporated, this strategy can be extended to achieve accurate recognition for more other objects in daily life. Movie S3 demonstrates utilizing the proposed tactile-visual fusion strategy to accurately recognize similarly-shaped crumpled paper, napkin, and plastic bag, which are easily confused by vision alone.

In order to demonstrate the superiority of the tactile-visual fusion recognition strategy, we utilize the tactile-visual fusion strategy to identify 10 items of daily life including paper, cleaning cloth, napkin, plastic bag, plastic bottle, orange peel, empty cup, cup with cold water, alcohol, and hot water (photographs in Fig. S14). For each item, we collect 70 samples, and we randomly divide the collected datasets into the training set, the validation set, and the test set (the ratio is 4:1:2). The model training takes about 0.33 s. We also compare the results using only vision or only tactile recognition. Here, the results shown in Fig. 5b–d are obtained from independent experiments using the corresponding recognition methods respectively. The recognition confusion matrix using only vision is shown in Fig. 5b, and the total recognition accuracy is only 59%. The misrecognition mostly occurs on shapeless and cup-shaped objects. For shapeless objects (e.g., crumpled print paper, napkin, or plastic bag), they lack a distinct shape and have similar colors, making them easy to confuse with each other using visual sensing. For cup-shaped objects, it is hard to determine the liquid content by vision due to the obstruction of the line-of-sight and transparency of the liquid. Using only tactile sensing to identify the above objects, the recognition confusion matrix is shown in Fig. 5c, and the total recognition accuracy reaches 92%. Tactile sensing can achieve high recognition accuracy on most objects. However, it is difficult for the tactile sensor to distinguish objects with complex shapes such as orange peel (75%). Furthermore, using the proposed tactile-visual fusion recognition strategy that combines the advantages of both touch and vision senses achieves the highest recognition accuracy of 96.5% (Fig. 5d). Combining vision also assists in the object position and pose for fine grasping.

### Robotic desktop-cleaning task for housekeeping assistance

Furthermore, we apply the proposed tactile-visual fusion robot to real-life scenarios, the robot autonomously accomplishes the desk-cleaning tasks (Movie S4 and S5). In this task, the robot coordinates all components (robotic arm, robotic hand, AGV, camera, and tactile sensors) based on the tactile-visual fusion architecture shown in Fig. 4a to execute various actions and finally clean up the desktop items, as

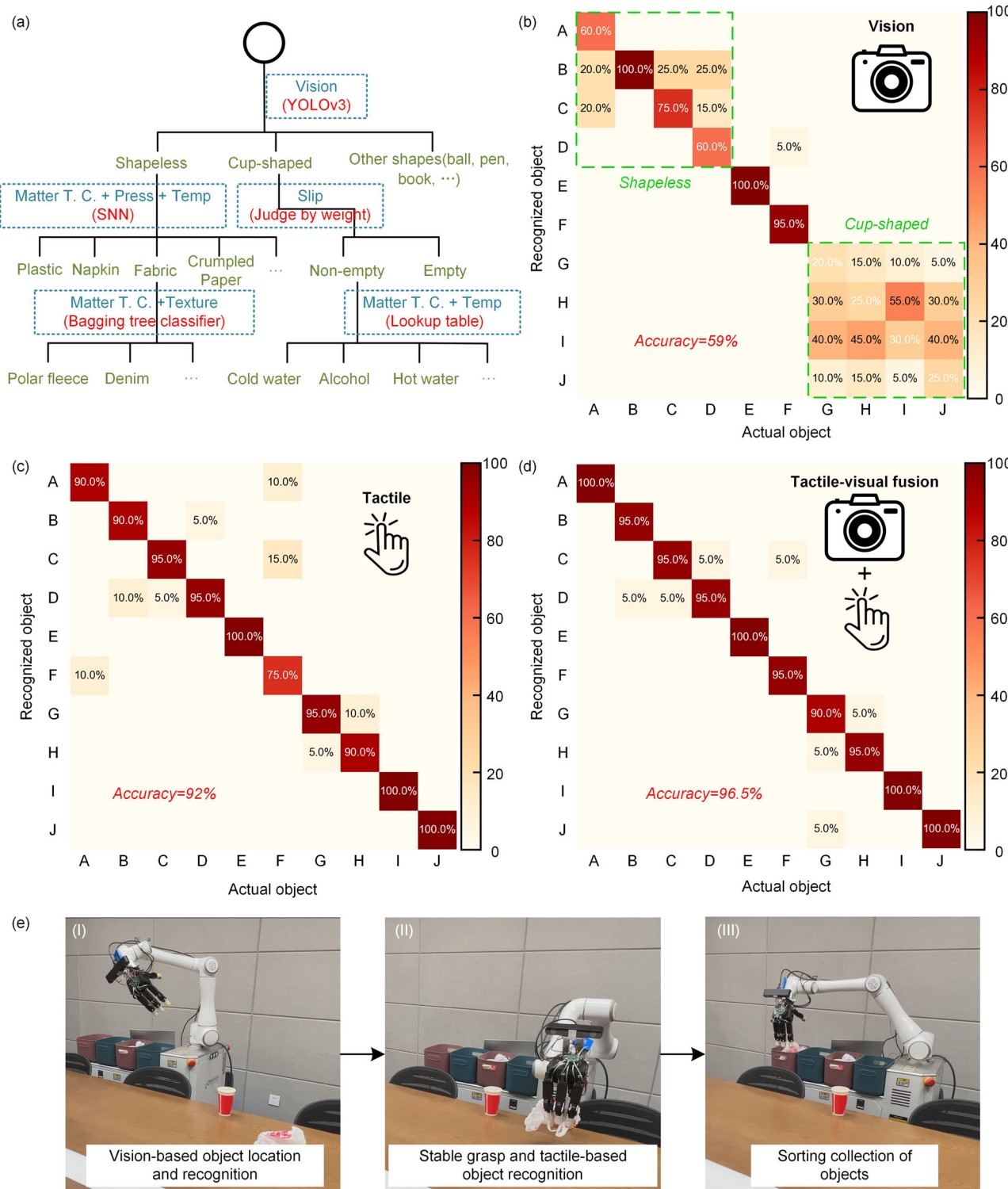

**Fig. 5 | Recognition strategy with tactile-visual fusion for object sorting and desk cleaning. a** The tactile-visual fusion recognition strategy, where Matter T. C., Press, and Temp refer to matter thermal conductivity, pressure, and temperature respectively. **b** The confusion matrix of object recognition using only vision, the total recognition accuracy is only 59%. **c** The confusion matrix of object recognition using only tactile sense, the total recognition accuracy is 92%. **d** The confusion matrix of object recognition using the tactile-visual fusion recognition strategy, the recognition accuracy reaches 96.5%. A = crumpled paper, B = cleaning cloth, C = napkin, D = plastic bag, E = plastic bottle, F=orange peel, G = cup with cold water, H = cup with alcohol, I = cup with hot water, J = empty cup. **e** Tactile-visual fusion robot helps a desk-cleaning task (Movie S4). (I) Vision-based object location. (II) Tactile-based stable grasping and object recognition. (III) Elaborate sorting and collection.

shown in Fig. 5e. Firstly, the robot enters the room, utilizes its camera to scan and locate the items on the table (Fig. 5e(I)), and moves to the vicinity of the items by AGV. And then the robot employs the tactile-visual fusion grasping strategy to stably grasp the objects (Fig. 5e(II)). At the same time, the robot identifies the types of objects using the tactile-visual fusion recognition strategy, and places these objects into the sorting boxes according to their catalogs (Fig. 5e(III)). Notably, when dealing with a cup containing liquid, the robot capably detects the liquid by tactile-based grasping, then pours the liquid into a water tank, and finally deposits the empty cup into the recyclable box (details in Movie S5). For some items that are difficult to grasp, such as a pen, a piece of paper, book, the robot with tactile-visual fusion can intelligently handle them by moving the objects to the edge of the table and then dexterously grasping them just like humans (Movie S4). This elaborate sorting and collection ensure the success of excellent housekeeping service.

## Discussion

In this paper, we propose a robotic flexible tactile sensor based on unified thermosensation, which enables multimodal perceptions of contact pressure, temperature, thermal conductivity, texture, and slippage. Thereinto, the slipping detection boasts an ultra-sensitivity with a low detection limit of 0.05 mm/s and an ultra-fast response time of 4 ms, far surpassing the performance of other sliding detection sensors. The good ability of slip detection ensures reliable robot grasping and stable holding to avoid objects dropping or endangering individuals. What's more, utilizing the thermal conductivity and texture perception ability, the fabric recognition achieves a high accuracy of 94.3%.

Visual recognition can identify objects with large differences in appearance, but it is difficult to distinguish visually similar objects such as napkin and cleaning cloth. Also, vision cannot recognize the transparency liquid in a cup. Although tactile recognition can well distinguish materials, but the recognition accuracy for objects with complex shapes such as orange peels is poor. In addition, due to the lack of visual guidance, the robot with only tactile sensing cannot accomplish the task such as object location, making it difficult to apply in real-life scenarios. Aiming to dexterously handle daily necessaries for housekeeping service, the robot needs to integrate tactile and visual sensing capabilities, while effectively coordinating them to accomplish the perception and cognition, strategy decisions, and system control. Therefore, we propose a tactile-visual fusion robot architecture integrating tactile and visual information from the signal level, perception level, and decision level, endowing the robot with robust sensing capabilities and execution proficiency. On this basis, we devise the corresponding tactile-visual fusion strategies for object grasping and object recognition. The grasping strategy utilizes fast and sensitive slip feedback to implement fine grasping with minimum grip strength, and the tactile-visual fusion recognition strategy employs a hybrid cascade strategy to implement accurate recognition of various daily necessities including identifying the liquid content in a cup. We apply the proposed recognition strategy to identify common objects of daily life, achieving a recognition accuracy of 96.5%, significantly surpassing only visual (59%) or only tactile (92%) recognition. Furthermore, utilizing the proposed tactile-visual fusion architecture and grasping/recognition strategies, a robot autonomously accomplishes a desktop-cleaning task. The results demonstrate the promising potential of smart robots with tactile-visual fusion for housekeeping applications, significantly reducing the need for manual labor. The developed multimodal tactile sensors and the proposed tactile-visual fusion robot architecture endow the robot with excellent perceptual and executive capabilities, facilitating flexible and reliable interaction with humans and helping humans in daily lives. In the future, we will consider utilizing more advanced algorithms such as large language model to further expand the capabilities of robots.

## Methods

### Fabrication of the tactile sensor

The tactile consists of a top sensing layer, a bottom sensing layer, a PDMS layer, and a porous material in the middle. The detailed fabrication process for each part are as follows:

The top sensing layer and the bottom sensing layer have totally the same structure and fabrication process. As Fig. S11a shows: (i)The sensing layer is fabricated on a flexible polyimide substrate. (AP8525R, DuPont Co. Ltd., Wilmington, America) (ii) Pads and wires are fabricated on polyimide substrate by flexible printed circuit board (FPCB) technology. (iii) Spin coat 30 μm thick photoresist (KXN5735-LO, Rdmicro Co. Ltd., Suzhou, China) on the polyimide substrate and use lithography to obtain the pattern. (iv) The chromium/platinum (35 nm/140 nm) films are sequentially deposited by sputtering. (v) Soak it in acetone for 2 h to remove the photoresist and form the corresponding pattern. (vi) Deposit parylene film with a thickness of 6 μm as a protective layer.

The tactile sensor is assembled by following steps: (i) A pure PDMS layer (the ratio of base agent: cross-linker was 10:1 wt%) is cured in a thin PMMA mold (14 mm × 10 mm × 2 mm) at 75 °C for 2 hours. (ii) Stack another thick PMMA mold (10 mm × 10 mm × 2 mm) on the pure PDMS layer and fabricate porous material as follows: The porous material is fabricated by mixing silver nanoparticles (diameter <100 nm, S110970, Aladdin Co. Ltd., Shanghai, China), prepared PDMS solution (Sylgard 184, Dow Corning Company, Wiesbaden, Germany, the ratio of base agent: cross-linker was 10:1 wt %), and citric acid monohydrate particles (CAM, Sinopharm Chemical Reagent Co. Ltd., Shanghai, China). The volume ratio of silver nanoparticles is 2.5 vol%. The mass ratio of PDMS: CAM is 1:3.5. After fully stirring the mixture (10 minutes), the mixture is cured at 75 °C for 3.5 h in PMMA mold (10 mm × 10 mm × 2 mm). Then the mixture is immersed in ethanol for 24 h to dissolve the CAM to form the porous material. Finally, it is washed with deionized water and dried at 70°C for 1 hour. (iii) After demolding and drying, adhere the sensing layers to the upper surface of the PDMS layer and the lower surface of the porous material. The corresponding fabrication process is shown in Figure S14b.

### Calibration of the tactile sensor

Pressure calibration is conducted by applying force on the tactile sensor using a mechanized z-axis stage (Handpi Co. Ltd., Yueqing, China) with a force gauge (SH-50, Sundoo Co. Ltd., Wenzhou, China, 0.01 N of resolution). It is worth mentioning that for the pressure detection limit, we place light objects on the sensor several times to see if the sensor responds. We record the mass of the items and calculate the detection limit of the sensor. Temperature responses of the sensor are tested in a temperature-controlled oven (OGH60, Thermo Fisher Scientific Co. Ltd., Massachusetts, America). We measure the sensor responses five times at 33–53 °C.

### Signal sampling and processing of the tactile sensor

The tactile signals ($U_p$, $U_{pc}$, $U_i$, and $U_{ic}$) are sampled by using an 18-bit analog-to-digital converter (AD7608, Analog Devices, Massachusetts, America) after being conditioned with the constant temperature difference circuit (shown in Figure S1) and a low-pass filter with a cutoff frequency of 678.6 Hz. Then the signals are packaged and sent to a laptop by using a microcontroller unit (STM32L476, STMicroelectronics, California, America) through a cluster communication port (COM port).

### Extraction of macro features and micro features

For the interface signal, we use 2000 points of smooth filtering to obtain the macro feature signal, and the rest are the micro feature signal. The macroscopic characteristic signal can directly reflect the slipping state of the object. The fundamental frequency ($f$) and its

higher harmonics can be obtained by performing fast Fourier transform (FFT) changes on the microscopic characteristic signal, which can be used to reflect the surface characteristics of the object. The above process is completed using MATLAB R2021b.

## Process of fabric recognition

We collect the top sensing signals of different fabrics when they pass across the sensor surface at a speed of 1 mm/s. It is then converted into macro features and micro features according to the previously mentioned method, which are used to reflect the thermal properties and texture of the fabric respectively. Using the bagging tree classification toolbox in MATLAB R2021b, the macroscopic features ($U_{macro}$) and the fundamental frequency ($f$) in microscopic features are used as inputs, and finally the recognition results are obtained.

## Configuration of the tactile-visual fusion robot

The robot mainly consists of robotic arm (EC66, ELITE Co. Ltd, Suzhou, China), robotic hand (Allegro hand, WONIK ROBOTICS, Seongnam-si, Korea), binocular depth camera (ZED 2i, Stereolabs Co. Ltd, San Francisco, America) and automated guided vehicle (Oasis-600C, STANDARD Co. Ltd, Shenzhen, China). The signals of the tactile sensors are sampled and transmitted to a PC via COM, and then processed to generate control commands by using the customized compliant control algorithm programmed in Python (Python 3.8.0). Then the commands are transmitted to the commercial robot controller-box via local area network.

## Hand-eye calibration and camera calibration

Taking account of the movement of the robot, the ZED 2i stereo camera is fixed to the end of the robot arm to ensure observation of the surroundings. Zhang's method[52] is used to obtain the intrinsic and extrinsic parameters of the left camera because the camera coordinate system is on the left camera for ZED 2i stereo camera. A calibration chessboard with $7 \times 10$ squares (each square is 20 mm × 20 mm) is printed and attached to a planar surface (as shown in Fig. S5). The robot arm equipped with a camera moves to 34 different orientations to take photos and record the robot arm parameters in every orientation. Knowing the robot arm parameters, intrinsic and extrinsic parameters of the camera, the following equations can be used to solve hand-eye calibration (solve $H_{cam}^{arm}$, the transformation between robot arm and the left camera) because the calibration chessboard is static in robot base system.

$$H_{cal\,i}^{cam} H_{cam\,i}^{arm} H_{arm\,i}^{base} = H_{cal\,j}^{cam} H_{cam\,j}^{arm} H_{arm\,j}^{base}$$

$H_{arm}^{base}$ is the transformation between the robot base system and robot arm, which can be obtained from robot arm parameters. $H_{cam}^{arm}$ is the transformation between the robot arm and stereo camera solved from hand-eye calibration. $1 \le i \le 34$, $1 \le j \le 34$. Written as the following equation of the form $AX = XB$, using Tsai's method to solve $H_{cam}^{arm}$[53].

$$H_{cam\,j}^{cal} H_{cal\,i}^{cam} H_{cam}^{arm} = H_{cam}^{arm} H_{arm\,j}^{base} H_{base\,i}^{arm}$$

Then, the objects can be located in the robot base coordinate system. $P_{cam}$ is the objects' position in the camera coordinate system obtained by the stereo camera.

$$P_{base} = H_{arm}^{base} H_{cam}^{arm} P_{cam}$$

## Details of the shallow neural network

We use a shallow neural network to utilize the tactile signals for recognition. The neural network has only one hidden layer with ten neurons, the input signals are voltages $U_i$ and $U_p$, and the output signals are recognition results.

## Data availability

The data that support the findings of this study are available within the paper and the Supplementary Information. The source data generated in this study are provided in the Source Data file. Source data are provided with this paper.

## Code availability

The MATLAB code for fabric recognition and the Python code for dexterous robotic housekeeping are available on GitHub at https://github.com/mq-0109/Multimodal-tactile-sensing-fused-with-vision-for-dexterous-robotic-housekeeping.

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

## Acknowledgements

The work was supported by the "National Natural Science Foundation of China" (No. 51735007, R.Z.) and "Beijing Natural Science Foundation" (Grant No. 3191001, R.Z.).

## Author contributions

R.Z. conceived the idea and directed the study. Q.M. fabricated the device. Q.M. and Z.J.L. designed and performed the experiments. Z.J.L., Q.M., J.F.Y., and R.Z. performed the visualization of the robot. Q.M. and R.Z. drafted the manuscript, and all authors contributed to writing and revising the manuscript. Q.M. and Z.J.L. contributed equally to this work.

## Competing interests

The authors declare no competing interests.
