## [Peer Review File · Nature Communications]

Reviewers' comments:

Reviewer #1 (Remarks to the Author):

In general, the manuscript is interesting. However, there are main issues that need to be addressed:

1- The lack of information necessary to replicate the results by other researchers. for example sensor fabrication. Also, I suggest authors share their codes and collected tactile and visual data,

2- The scientific contribution of this paper is not clear to me. The authors argue that

“ we propose a tactile-visual fusion robot architecture combining tactile and visual fusion sensing with an intelligent decision system...”. I could not find neither the novel architecture for data fusion, nor understand the “intelligent decision system” from the current version of the manuscript.

3- What is the scalability of the proposed method?

Other main points:

1- The fabrication process of the multimodal tactile sensor is not clear enough. The current description is general and detailed information should be provided.

2- The experiments are simple for this journal. In Fig.3, the authors only used 8 fabrics which is not enough. They should test at least 10 different fabrics. Also, they should test their sensor with a combination of fabrics. For example, the first half from one fabric and the second half from another fabric to see if the texture changes in one sample, how does this sensor respond?

3- The authors argue “For the perception level, the computer transforms the sensor signals into corresponding cognitions”. It was not clear how this transformation happened.

4- It would have been nice to read commentary and expectations on the objects selected by the authors. Which ones are visually difficult to recognize? Which ones are difficult with tactile information only? Which ones would benefit from information fusion?

5- They used “shallow neural network (SNN)” and explained it in 2 lines (Supplementary). The contribution of this network was not discussed in this data fusion architecture.

6- How was the data split into training/validation/test subsets? How long did it take to train the model?

7- How does the performance or accuracy change if other models were chosen as the base model, e.g. VGG16?

Minor points:

8- The discussion is short! The merits and limitations of the different existing solutions should be discussed with the current approach.

9- Fig. 1 is confusing and needs more clarification. It should be modified to be more relevant to the main goals of the paper.

Reviewer #2 (Remarks to the Author):

This paper seems to catch up a trend in the field of robots and sensors by combining tactile and visual sensory with AI for the dexterous manipulation.

I carefully read the manuscript with my interest.

The followings need to be addressed for the publication

1. page 4, line 17~19 and Fig. 2(a), The notations for the hot-film and cold-film in the texts and Fig. 2(a) are different. Descriptions in the main text seem to be right.

2. Page 5, line 8, The evidence seems to be not enough for claiming the sensor performance in terms of a low detection limit to be 0.01 N.

3. page 5, line 14 what does "CTD" stand for?

4. Fig. 2g, How the sensor can discriminate the slip from the actual contact pressure increase (or decrease)?

I think it wouldn't be capable of independent sensing with this device structure and the sensing principle.

5. page 5, line 25, The evidence seems to be not enough for claiming the tactile sensor is immune from temperature variation.

There are no descriptions about the symbol of U_c , U_p , R_a , R_b , R_t in the supplementary material.

6. The circuit diagram in Figure S1 needs to be corrected. In the diagram, the output of a differential amplifier is connected to the input of Wheatstone bridge circuit.

7. Dimensions (or size) of the devices must be described in the text and indicated in Fig. 2.

8. What if the contact area is smaller than the sensing area of the sensor? The output characteristics is subject to change according to the area of contact.

Reviewer #3 (Remarks to the Author):

This work proposes an interesting thermometry sensor to perceive pressure, slipping, temperature, thermal conductivity, and surface texture for use with robotic systems. It further utilises this sensor along with visual data to present an approach for allowing a robot to undertake complex tasks such as handling a cup containing liquid.

The work outlines that there is a need to enhance current robotic systems (and multimodal sensing), and the need for robots to include tactile sensors over using purely visual systems. While the general framing of the problem is good I am still uncertain where the presented work sits within the state-of-the-art. It is not clear, as written, if the proposed sensor is a novel development, or if the proposed novelty of the

work is the sensors fusion (where again, the exact novelty is not clear). Some sections in the introduction make statements without a supporting reference. The other references in the paragraph may support these assertions, but as written this is unclear. For example, '...for intricate robotic tasks, sole tactile perception faces challenges such as a lack of visual guidance and an inability to locate objects in the environment.'

The data from the sensor itself is impressive, however the repeatability of the measurements is unclear. The text frequently presents values without errors (page 5 for example lists values such as the low detection limit without errors). I could not find reference to repeat measurements in the Materials and Methods section either. Figure 2f has error bars, implying that repeat measurements were taken, however Figure 2c and 2e do not. While I believe that the sensor works, which is evident later in the manuscript, it is very difficult to draw meaningful conclusions regarding the sensor without information on the repeatability of individual sensors and between different sensors. Generally I felt that the Materials and Methods section required further details, for example the source of the fabrics used.

I have a further question regarding the sensor itself. This is described as a pressure sensor, yet the only data presented relates to force sensing. Would it be correct to assume that the sensitivity to force only was measured? As mentioned above, the materials and methods section lacks certain details, so I am uncertain how this characterisation (specifically for Fig. 2c) was conducted.

Overall further details would be required for someone to repeat the presented work. Without this further information it is challenging to fully comment on whether the methodology used is fully sound.

Dear Editor and Reviewers,

Thank you very much for your comments and efforts contributed to our paper. We have carefully investigated all comments. They are very helpful to improve the manuscript. We believe have led to the substantial revision of the paper. The responses to these comments are listed in the following and all of the revisions have been marked in red font in the text of the revised Manuscript and Supplementary Materials.

COMMENTS TO AUTHOR:

Reviewer #1

In general, the manuscript is interesting. However, there are main issues that need to be addressed:

Comment #1

The lack of information necessary to replicate the results by other researchers. for example sensor fabrication. Also, I suggest authors share their codes and collected tactile and visual data,

Response #1

Thanks very much for your comments on our work.

We have introduced the fabrication process of the tactile sensor in the Materials and Methods section of the Supplementary Materials (Page 2). This section describes the fabrication of sensing layers, porous materials, and assembly of tactile sensors. We have added Figure S15 to show the fabrication process. We also have added the sensor calibration process in the Materials and Methods of the Supplementary Materials. These fabrication process and sensor calibration process ensure replicating the results of the sensor.

As for the code and the tactile and visual data collected in this manuscript, we are glad to share them with other researchers.

The corresponding contents can be seen on Page 2 and 21 in the revised Supplementary Materials.

Comment #2

The scientific contribution of this paper is not clear to me. The authors argue that “we propose a tactile-visual fusion robot architecture combining tactile and visual fusion sensing with an intelligent decision system...”. I could not find neither the novel architecture for data fusion, nor understand the “intelligent decision system” from the current version of the manuscript.

Response #2

Thanks for your comment.

Our proposed tactile-visual fusion strategy is a novel decision-level fusion rather than simple data-level fusion for dexterously grasping and accurately recognizing various objects of daily life. The tactile-visual fusion strategy is utilized for grasping various objects (Figure 4b), where the vision assists in determining object position and pose, and the slip-based tactile feedback control ensures dexterous robotic grasping. Besides, we propose a novel cascade strategy (Figure 5a) where visual and tactile senses cooperate with each other by using a tailor-designed cascade classifier to achieve accurate recognition on daily necessities. Our tactile-visual fusion strategy is superior over existing works, by utilizing multimodal tactile sensing fused with vision to achieve dexterous robotic grasping and accurate recognition on daily objects.

To validate our proposed tactile-visual fusion architecture and the decision system, besides the validations in grasping objects shown in Figure 4c-4f and the daily item recognition shown in Figure 5d, we provide a showcase of robotic desk-cleaning tasks, the robot autonomously accomplishes multi-item sorting and cleaning desktop, handling challenging tasks, such as sorting out the paper-cup containing liquid and picking up a piece of paper, etc., demonstrating the promising potential of smart robots for challenging housekeeping applications. We provide movies (Movie S1~S4 and an added Movie S5) to demonstrate our accomplishments. These demonstrations fully validate the effectiveness of the tactile-visual fusion architecture and robot decision system.

The corresponding contents can be seen on Page 11-14 in the revised Manuscript.

Comment #3

What is the scalability of the proposed method?

Response #3

Thanks for your comment.

Our proposed tactile-visual fusion method is highly scalable. We split the robot's functions into multiple modules such as visual recognition and location, object surface perception, slip detection, and tactile recognition. Each module can not only accomplish its function independently, but also combine with each other to complete more complex tasks. Here, we provide one more house-keeping example to showcase the scalability of our method in Movie S5.

Comment #4

The fabrication process of the multimodal tactile sensor is not clear enough. The current description is general and detailed information should be provided.

Response #4

Thanks for your suggestion.

We have added a more detailed sensor fabrication process in the Materials and Methods

section, and also supplemented Figure S15 for clearly presenting the fabrication process. The corresponding revision can be seen on Page 2 and 21 in the revised Supplementary Materials.

Comment #5

The experiments are simple for this journal. In Fig.3, the authors only used 8 fabrics which is not enough. They should test at least 10 different fabrics. Also, they should test their sensor with a combination of fabrics. For example, the first half from one fabric and the second half from another fabric to see if the texture changes in one sample, how does this sensor respond?

Response #5

Thanks for your suggestion.

Different fabrics have different thermal conductivities and surface textures, which can be detected by our tactile sensor. Therefore, we use the tactile sensor for fabric identification. We have added two additional fabrics (Figure S7 and Figure S8) for fabric recognition, and the recognition accuracy reaches 94.3% (Figure 3i), proving the effectiveness of this method. In addition, we also added the tactile response when using a combination of two fabrics (Figure 3h). It is seen that when two fabrics are spliced together, the sensor capably contacts two fabrics in sequence during a slipping process, and the corresponding macroscopic and microscopic characteristics of the sensor signal can be utilized to identify the fabrics in sequence.

The corresponding revision can be seen on Page 7-8 in the revised Manuscript and Page 13-14 in the revised Supplementary Materials.

Comment #6

The authors argue “For the perception level, the computer transforms the sensor signals into corresponding cognitions”. It was not clear how this transformation happened.

Response #6

Thanks for your comment.

After the robot collects visual signals and tactile signals, the computer combines them with corresponding algorithms to transform them into cognitive information (e.g., decide to move or not, grab, recognize, or sort objects, etc.) for the robot (Figure 4a). Specifically, the visual signals combined with the YOLOv3 algorithm can help the robot recognize the surrounding environment, implement object location, and recognize objects based on their shape, size, color, etc., resulting in the categories such as ball-shaped, bottle-shaped, cup-shaped, shapeless, etc. (Figure 4a and Figure 5a); the tactile signals combined with the feature extraction algorithm can perceive the pressure, sliding, thermal conductivity of the object, temperature, etc., which are used for the robot to implement dexterous grasping object by a slip feedback control and achieve object recognition by a shallow neural network (SNN) (Figure 5a).

The corresponding contents can be seen on Page 9-14 in the revised Manuscript.

Comment #7

It would have been nice to read commentary and expectations on the objects selected by the authors. Which ones are visually difficult to recognize? Which ones are difficult with tactile information only? Which ones would benefit from information fusion?

Response #7

Thanks for your suggestion.

As we have mentioned on Page 12, visual information is difficult to distinguish between crumpled paper, cleaning cloth, napkin, and plastic bag since they have similar colors and shapes (photographs shown in Figure S14). In addition, the transparent liquid contained in a cup and the object temperature cannot be recognized or detected by vision. The specific recognition result (accuracy of 59%) can be seen in Figure 5b.

For tactile information, it is difficult to distinguish the objects with complex shapes such as orange peel. As shown in Figure 5c, the recognition accuracy for the orange peel is 75%, much lower than other objects.

Therefore, the tactile-visual fusion (Figure 5a) enables the robot to accurately recognize visually-similar items (crumpled paper, cleaning cloth, napkin, and plastic bag) and cup-shaped items containing liquid (cup with cold water, cup with alcohol, cup with hot water, and empty cup), and items with complex shapes (orange peel), which greatly makes up for the shortcomings of only visual or only tactile method.

The corresponding contents and revision can be seen on Page 12-14 in the revised Manuscript.

Comment #8

They used “shallow neural network (SNN)” and explained it in 2 lines (Supplementary). The contribution of this network was not discussed in this data fusion architecture.

Response #8

Thanks for your suggestion.

We have demonstrated the role of the shallow neural network (SNN) in the tactile-visual fusion recognition strategy in Figure 5a. We collect the tactile sensing signals (U_i from the top sensing layer and U_p from the bottom sensing layer) of shapeless objects, and use them to train a neural network with a single hidden layer containing ten neurons. The trained network is used to distinguish the material, hardness, and temperature of objects, such as plastic, napkin, fabric, etc.

The corresponding contents can be seen on Page 14 in the revised Manuscript.

Comment #9

How was the data split into training/validation/test subsets? How long did it take to

train the model?

Response #9

Thanks for your suggestion.

We have supplemented the information of splitting the training/validation/test datasets and the training time of the model in the revised manuscript. For each item, we collect 70 samples, and we randomly divide the collected datasets into the training set, the validation set, and the test set (the ratio is 4:1:2). Figure 5 shows the recognition results of the test set.

The average time taken for the model training is about 0.33 s.

The corresponding revisions can be seen on Page 13 in the revised Manuscript.

Comment #10

How does the performance or accuracy change if other models were chosen as the base model, e.g. VGG16?

Response #10

Thanks for your suggestion.

In fact, target detection algorithms include VGG16, YOLOv3 (we used in this manuscript) and many other algorithms. According to literature (*Sensors* **2020**, 20, 1678, doi:10.3390/s20061678; *2021 6th International Conference for Convergence in Technology (I2CT)*. IEEE, **2021**, doi: 1-810.1109/I2CT51068.2021.9417895), there is no significant difference in the recognition accuracy between VGG16 and YOLOv3, except that the frames per second (FPS) of YOLOv3 is higher. It is foreseeable that using the VGG16 model can achieve a recognition accuracy similar to the YOLOv3 model, which also demonstrates the effectiveness of the tactile-visual fusion method we proposed. Of course, we will consider comparing VGG16 with YOLOv3 in our future work.

Minor Comment #1

The discussion is short! The merits and limitations of the different existing solutions should be discussed with the current approach.

Response #1

Thanks for your suggestion.

We have carefully revised the discussion and added the merits and limitations of the different solutions. Visual recognition can identify objects with large differences in appearance, but it is difficult to distinguish visually-similar objects such as napkin and cleaning cloth. Also, it cannot recognize the transparent liquid in the cup. Tactile recognition can well distinguish materials, but the recognition accuracy for objects with complex shapes such as orange peels is poor. In addition, due to the lack of visual guidance, the robot with only tactile sensing cannot accomplish the task such as object

location, making it difficult to apply in real-life scenarios. The tactile-visual fusion strategy proposed in this article complements each other's advantages. We propose the tactile-visual fusion robot architecture integrating tactile and visual information from the signal level, perception level, and decision level, endowing the robot with robust sensing capabilities and execution proficiency. The robot can accomplish object location, stable grasping, and accurate recognition. In virtue of the proposed method, the robot can autonomously accomplish the desktop-cleaning tasks, the robot autonomously accomplishes multi-item sorting and cleaning desktop. In the future, we will consider utilizing more advanced algorithms such as large language model to further expand the capabilities of robots.

The corresponding revisions can be seen on Page 15-16 in the revised Manuscript.

Minor Comment #2

Fig. 1 is confusing and needs more clarification. It should be modified to be more relevant to the main goals of the paper.

Response #2

Thanks for your suggestion.

We have modified Figure 1 to make it more relevant to the topic of this article. Also, we have modified the corresponding text to help readers better understand the topic. This manuscript reports great advancements in multimodal tactile sensing, tactile-visual fusion architecture, grasping control, and object recognition, as well as important application (housekeeping), which is a great advance.

The corresponding revisions can be seen on Page 3-4 in the revised Manuscript.

Thanks again for your valuable comments and suggestions.

Reviewer #2

This paper seems to catch up a trend in the field of robots and sensors by combining tactile and visual sensory with AI for the dexterous manipulation. I carefully read the manuscript with my interest. The followings need to be addressed for the publication.

Comment #1

Page 4, line 17~19 and Fig. 2(a), The notations for the hot-film and cold-film in the texts and Fig. 2(a) are different. Descriptions in the main text seem to be right.

Response #1

Thanks for your comments and suggestion.

We have modified the notations in Figure 2a. As described in the main text, the inner thermistor has a low resistance ($\sim 50 \Omega$, denoted as hot-film), while the outer thermistor has a higher resistance ($\sim 500 \Omega$, denoted as cold-film).

The corresponding revisions can be seen on Page 6 in the revised Manuscript.

Comment #2

Page 5, line 8, The evidence seems to be not enough for claiming the sensor performance in terms of a low detection limit to be 0.01 N.

Response #2

Thanks for your suggestion.

We have added an inset in Figure 2c to illustrate that the sensor has a low detection limit of 0.01 N.

The corresponding revisions can be seen on Page 6 in the revised Manuscript.

Comment #3

Page 5, line 14 what does "CTD" stand for?

Response #3

Thanks for your comment.

The CTD circuit (Figure S1) refers to the constant temperature difference circuit which ensures the temperature difference between the hot-film and the cold-film (environment temperature sensor) is constant and thus the tactile sensor is not affected by the ambient temperature. The details are in Supplementary text, Figure S1, and Figure S2.

The corresponding contents and revisions can be seen on Page 5 in the revised Manuscript and Page 5-8 in the revised Supplementary Materials.

Comment #4

Fig. 2g, How the sensor can discriminate the slip from the actual contact pressure increase (or decrease)? I think it wouldn't be capable of independent sensing with this device structure and the sensing principle.

Response #4

Thanks for your comment.

The slipping and the contact pressure are detected independently by the top and bottom sensing layers respectively. The sensor contains a top sensing layer (used to perceive slip, thermal conductivity, and object temperature) and a bottom sensing layer (used to perceive pressure and environmental temperature). The crosstalk between the top and bottom sensing layer is small. We have supplemented a response result of the top sensing layer under different pressures (loading and unloading) in Figure S3. You can see that the crosstalk is less than 4.7%, which shows that the contact pressure hardly affects the slip detection of the top sensing layer.

The corresponding revisions can be seen on Page 9 in the revised Supplementary Materials.

Comment #5

page 5, line 25, The evidence seems to be not enough for claiming the tactile sensor is immune from temperature variation. There are no descriptions about the symbol of U_c , U_p , R_a , R_b , R_t in the supplementary material.

Response #5

Thanks for your comment.

As mentioned above, the tactile sensor is immune from temperature variation. The temperature compensation is described in Supplementary Text of Supplementary Materials. We have supplemented the response results of the sensor (U_i and U_p) at different temperatures in Figure S2 in the Supplementary Materials. In the range of 25~45°C, the temperature effect on the the sensor (U_i and U_p) is less than 0.6%, indicating that the tactile sensor is immune from the temperature variation.

For the symbols of U_c , U_p , R_a , R_b , and R_t , we have added the descriptions in the supplementary text. And they refer to the corresponding symbols shown in Figure S1. The corresponding revisions can be seen on Page 5 and 7-8 in the revised Supplementary Materials.

Comment #6

The circuit diagram in Figure S1 needs to be corrected. In the diagram, the output of a differential amplifier is connected to the input of Wheatstone bridge circuit.

Response #6

Thanks for your comment.

Yes, the output of a differential amplifier is connected to the input of Wheatstone bridge

circuit to build a closed-loop feedback control circuit shown in Figure S1, which ensures the self-sustaining constant temperature difference between the hot-film and cold-film. That is the reason why the tactile sensor is immune from the environment variation.

The corresponding revision can be seen on Page 7 in the revised Supplementary Materials.

Comment #7

Dimensions (or size) of the devices must be described in the text and indicated in Fig. 2.

Response #7

Thanks for your suggestion.

The size of the tactile sensor is indicated in Figure 4a. In addition, we have added a description of the sensor size and the corresponding scale bar in Figure 2a. The radii of the hot-film and cold-film are 0.95 mm and 3.2 mm, respectively.

The corresponding contents and revisions can be seen on Page 4, 6 and 11 in the revised Manuscript.

Comment #8

What if the contact area is smaller than the sensing area of the sensor? The output characteristics is subject to change according to the area of contact.

Response #8

Thanks for your comment.

Yes, as you stated if the contact area is smaller than the sensing area of the sensor, the output characteristics are subject to change according to the area of contact. Therefore, scaling down the sensing area of the sensor would help to eliminate the contact area effect, which will be considered in our future work.

Thanks again for your valuable comments and suggestions.

Reviewer #3

This work proposes an interesting thermometry sensor to perceive pressure, slipping, temperature, thermal conductivity, and surface texture for use with robotic systems. It further utilises this sensor along with visual data to present an approach for allowing a robot to undertake complex tasks such as handling a cup containing liquid.

Comment #1

The work outlines that there is a need to enhance current robotic systems (and multimodal sensing), and the need for robots to include tactile sensors over using purely visual systems. While the general framing of the problem is good I am still uncertain where the presented work sits within the state-of-the-art. It is not clear, as written, if the proposed sensor is a novel development, or if the proposed novelty of the work is the sensors fusion (where again, the exact novelty is not clear).

Response #1

Thanks for your comments.

In fact, the multimodal tactile sensor and the sensor fusion method proposed in this article are both important innovations. First, the tactile sensor proposed in this manuscript implements **multimodal perceptions of pressure, temperature, thermal conductivity, texture, and slippage**. To the best of our knowledge, such a multimodal integrated tactile sensor has never been reported and achieves superior sensing performances over other existing sensors. Importantly, the proposed tactile sensor is endowed with an **ultrasensitive (0.05 mm/s) and ultrafast response (4 ms)** to slip detection which is extremely indispensable for fine grasping control as shown in Figure 3b, 3c, 4c~f. The texture detection capability enables the sensor fine recognize daily necessities, such as fabrics, as shown in Figure 3g~3i. These perceptual capabilities are important for assisting robots in working in complex house-living environments.

Furthermore, we propose a novel **tactile-visual fusion robot architecture** that can autonomously accomplish object location, stable object grasping, and object recognition. The proposed **tactile-visual fusion grasping strategy** is utilized for grasping various objects, where vision assists in determining object position and pose, and slipping-based tactile feedback control ensures dexterous robotic grasping. In addition, the proposed **tactile-visual fusion recognition strategy** using a tailor-designed cascade classifier achieves accurate recognition of a variety of objects, which is significantly superior to a single tactile or single visual recognition method.

In application, we showcase a robotic desktop-cleaning task, for the first time, the robot autonomously accomplishes multi-item sorting and cleaning desktop, demonstrating its promising potential for smart housekeeping.

In summary, compared with other existing works, this paper has the following distinct innovations:

1. **New multimodal sensing.** The proposed tactile sensor capably perceives contact pressure, temperature, thermal conductivity, texture, and slippage simultaneously.

2. **Superior sensing performance.** The slip detection of the tactile sensor achieves an extremely sensitive detection limit (0.05 mm/s) and extremely fast response time (4 ms), which is indispensable for stable and reliable grasping control, mitigating the risk of objects dropping or endangering individuals.
3. **New tactile-visual fusion robot architecture.** We propose a tactile-visual fusion robot architecture that seamlessly encompasses multimodal senses at the bottom level to robotic decisions at the top level, empowering robots with the capability to interact effectively with intricate environments.
4. **New stable grasping strategy.** Utilizing the ultrasensitive and ultrafast slip detection of the sensor, we propose a slip-feedback grasping strategy, which achieves stable grasping with minimal grip strength to avoid crushing fragile objects. This grasping strategy can be generalized to grab any object.
5. **New tactile-visual fusion recognition strategy.** We propose a new tactile and visual fusion strategy to achieve accurate recognition of various daily objects, which is significantly better than single vision or single touch.
6. **New housekeeping application.** We employ the robot for a desktop cleaning task (Movie S1). The robot autonomously accomplishes multiple tasks such as object location, grasping, recognition, and sorting, demonstrating the promising potential of smart robots for housekeeping applications.

Comment #2

Some sections in the introduction make statements without a supporting reference. The other references in the paragraph may support these assertions, but as written this is unclear. For example, '...for intricate robotic tasks, sole tactile perception faces challenges such as a lack of visual guidance and an inability to locate objects in the environment.'

Response #2

Thanks for your suggestion.

We have carefully examined the introduction and added corresponding references to make the introduction clearer and easier to understand.

The corresponding contents and revisions can be seen on Page 2-3 in the revised Manuscript.

Comment #3

The data from the sensor itself is impressive, however the repeatability of the measurements is unclear. The text frequently presents values without errors (page 5 for example lists values such as the low detection limit without errors). I could not find reference to repeat measurements in the Materials and Methods section either. Figure 2f has error bars, implying that repeat measurements were taken, however Figure 2c and 2e do not. While I believe that the sensor works, which is evident later in the

manuscript, it is very difficult to draw meaningful conclusions regarding the sensor without information on the repeatability of individual sensors and between different sensors. Generally I felt that the Materials and Methods section required further details, for example the source of the fabrics used.

Response #3

Thanks for your comments and carefully check.

We have added the error bars for the measurements in Figure 2-3. And we also have added the repeatability test of the sensor (Figure S4). Under 1000 cycle tests, the output of the sensor remains stable, indicating the good reliability and stability of the sensor. In addition, we also supplement the pressure response (0.01~20 N) and temperature response (33~51°C) of the sensor, and add the error bars in the Figure 2c and 2e.

We also add the information of the test equipment in the Materials and Methods. Pressure calibration is conducted by applying the force on the tactile sensor using a mechanized z-axis stage (Handpi HLD) with a force gauge (Sundoo SH-50). Temperature response of the sensor are tested in a temperature-controlled oven (OGH60, Thermo Fisher Scientific Co. Ltd.).

As for the source of the fabrics used, the fabrics are bought from the market.

The corresponding contents and revisions can be seen on Page 6 in the revised Manuscript and Page 2 and 10 in the revised Supplementary Materials.

Comment #4

I have a further question regarding the sensor itself. This is described as a pressure sensor, yet the only data presented relates to force sensing. Would it be correct to assume that the sensitivity to force only was measured? As mentioned above, the materials and methods section lacks certain details, so I am uncertain how this characterisation (specifically for Fig. 2c) was conducted.

Response #4

Thanks for your comments.

In robot applications, the pressure sensor is often utilized for the force measurement. Therefore, we utilize the force as the gauge to calibrate the pressure sensor. Generally, the pressure can be calculated by dividing the force by the sensing area. And the force calibration is conducted by applying force on the tactile sensor using a mechanized z-axis stage with a force gauge. The force sensing result of the pressure sensor is shown in Figure 2c.

The corresponding revisions can be seen on Page 6 in the revised Manuscript and Page 2 in the revised Supplementary Materials.

Comment #5

Overall further details would be required for someone to repeat the presented work.

Without this further information it is challenging to fully comment on whether the methodology used is fully sound.

Response #5

Thanks for your comment and suggestion.

We have added more detailed descriptions about sensor structure, fabrication, and measurements in the revised Materials and Methods section of Supplementary Materials, and carefully checked and modified the relevant pictures and text descriptions in the manuscript. In addition, we also have added a detailed schematic diagram of the sensor fabrication process (Figure S15) and a text description (Materials and Methods), as well as the sensor calibration process (Materials and Methods). We hope this supplementary information helps readers reproduce our work, and also help you better evaluate the research methods of this manuscript.

The corresponding revisions can be seen on Page 2-3 and 6 in the revised Manuscript and Page 2 and 21 in the revised Supplementary Materials.

All of the above revisions have been marked in red font in the text of the revised Manuscript and revised Supplementary Materials.

Thanks for the comments again.

Sincerely

REVIEWER COMMENTS

Reviewer #2 (Remarks to the Author):

Thank you for your responses to my comments.

Some are addressed, but there are still a few comments that need rigorous explanation.

Questions about responses on my previous comments.

1. Response #2

-> Adding an inset in Figure 2c is not enough. The electronics for signal processing would be one of critical aspects to determine the low detection limit. However, there is no detailed information or descriptions about the electronics. The inset in Figure 2c shows that the output level change is on the order of 1 mV for the 0.01 N force step with little noise. What kind of electronics could measure 1 mV level voltage with such a little noise? Do you use low-pass filter to reduce noise? Then what would be the bandwidth of the electronics? Is this bandwidth enough to be used for the control?

2. Response #4

 I think the independent sensing would be possible in a controlled or well-defined grasping situation. If someone is pouring cold or hot water into a cup which a robotic hand is holding, can the robot perceive the grasping situation correctly?

How the top sensing layer could discriminate slip, object temperature, and thermal conductivity originated from the same thermal flow changes?

3. Response #5

 The description on resistors R_a , R_b , and R_t is not still enough. R_a , R_b and R_t are resistors out of the device? What kind of resistors are used? what are their resistance values? How can you assume that all resistors are in the same ambient temperature?

4. Response #6

 Basically, the effect of temperature variation is minimized if all four active strain gauges form the Wheatstone bridge circuit. Most strain-gauge load cells use this principle. In the suppl. material, authors

claims that the temperature difference between hot and the environment is constant, but no evidence and explanation about the temperature independence about R_t is not provided. According to the last equation in the suppl. mater., ΔT is proportional to R_t . I also couldn't understand how the Wheatstone bridge circuit in Figure S1 could be balanced. R_h is about 50Ω and R_c is about 500Ω . R_t is connected to R_c , then what would be resistance values of R_a and R_b ?

5. Response #8

 Just scaling down the sensing area couldn't be a solution. In this case, the object could touch the other part of the sensing area. Actually, we can call the sensing device as "tactile sensor" if and only if the device has an array of sensing elements that could measure physical quantities upon touch over an area. If not, what is difference between "tactile sensor" and force sensor?

Reviewer #3 (Remarks to the Author):

This manuscript details an interesting thermometry sensor that is used to perceive pressure, slipping, temperature, thermal conductivity, and surface texture for use with robotic systems. The sensor is further utilised in combination with visual data to present an approach for allowing a robot to undertake complex tasks.

I think that the revised manuscript is much improved, and my main issues, the lack of detail regarding aspects of the methodology and (critically) the lack of detail regarding the repeatability of measurements have been addressed to some extent. I did note that full company details were not provided in most cases (city and country of supplier). While I am happy to see the addition of error bars to Figure 2c and 2e, I am still not certain how many repeat measurements were taken to get these measurements (regarding temperature response, it just says that it was tested multiple time). Is the presented data representative for one specific sensor, with measurements repeated, or multiple sensors? How were the error bars calculated? This information needs to be present in the manuscript.

My other remaining minor criticism relates to the framing of the novelty of the work. I highlighted this in my previous review, and the authors have provided an excellent and detailed explanation of why the work is novel, however I still feel that in the manuscript this is somewhat unclear. I think that three or four extra lines in the introduction clearly expressing this would make the novelty of the manuscript clearer to the reader.

Dear Editor and Reviewers,

Thank you very much for your comments and efforts contributed to our paper. We have carefully investigated all comments. They are very helpful to improve the manuscript. We believe this has led to the substantial revision of the paper. The responses to these comments are listed in the following and all of the revisions have been marked in red font in the text of the revised Manuscript and Supplementary Materials.

COMMENTS TO AUTHOR:

Reviewer #2

Thank you for your responses to my comments. Some are addressed, but there are still a few comments that need rigorous explanation. Questions about responses on my previous comments.

Comment #1

Adding an inset in Figure 2c is not enough. The electronics for signal processing would be one of critical aspects to determine the low detection limit. However, there is no detailed information or descriptions about the electronics. The inset in Figure 2c shows that the output level change is on the order of 1 mV for the 0.01 N force step with little noise. What kind of electronics could measure 1 mV level voltage with such a little noise? Do you use low-pass filter to reduce noise? Then what would be the bandwidth of the electronics? Is this bandwidth enough to be used for the control?

Response #1

Thanks for your comments.

We have supplemented a description of the electronics of the tactile sensor in the section of Materials and Methods. Here, we use an 18-bit sampling ADC (AD7608), which has a resolution of $\frac{5V}{2^{18}} = 0.019 \text{ mV}$ in the range of 0~5 V. In addition, we use a low-pass filter with a cutoff frequency of 678.6 Hz to reduce the noise in the circuit, so the signal shown in the inset of Figure 2c has a high signal-to-noise ratio. This bandwidth can fully meet the robotic control. For example, Movie S2 shows the tactile feedback control enabling the stable grasp of a cup with water being poured. The corresponding revisions can be seen on Page 17 in the revised Manuscript.

Comment #2

I think the independent sensing would be possible in a controlled or well-defined grasping situation. If someone is pouring cold or hot water into a cup which a robotic hand is holding, can the robot perceive the grasping situation correctly?

How the top sensing layer could discriminate slip, object temperature, and thermal

conductivity originated from the same thermal flow changes?

Response #2

Thanks for your comments.

We have provided a Movie S2 which shows your mentioned case. A person is pouring water into a cup which a robotic hand is holding, the tactile sensor detects the slip induced by the water weighting, and the robotic hand rapidly controls the grasping force to ensure a stable grasp. In virtue of the ultrasensitive and ultrafast slip sensing of the tactile sensor as well as the proposed tactile feedback control, the robot ensures dexterous and reliable grasping control to avoid dropping objects. You see our tactile sensing and control is still effective in complex unwell-defined grasping situations.

As for how to discriminate slip, object temperature, and thermal conductivity from the sensor signals, we have explained it in detail on Page 5 and 7 in the Manuscript. As we have mentioned on Page 5 in the main text and Page 2 in the Supplementary materials (equation (5)), the cold-film in the top sensing layer is utilized to detect the object temperature. Specifically, there is a good linear relationship between the cold-film voltage ratio of the top sensing layer $\eta_{object} = U_{ic}/(U_i - U_{ic})$ and the object temperature as shown in Figure 2e. The hot-film in the top sensing layer is utilized to detect the slip and the thermal conductivity, which is independent of the object temperature sensing (shown in Figure S2). As we have stated on Page 7, first the robotic hand comes into contact with the object, and the steady hot film signal in the top sensing layer is utilized to determine the thermal conductivity of the object as shown in Figure 2f. And then, the robotic hand grabs the object, picks it up, and holds it. During this grasping process, the top hot-film sensing signal combined with the pre-known thermal conductivity is utilized to determine the slipping state of the object as shown in Figure 3a, and is utilized for the rapid slip feedback control. You see the object temperature, thermal conductivity, and slip are detected discriminably.

The corresponding content and revisions can be seen on Page 5-8 in the revised Manuscript.

Comment #3

 The description on resistors R_a , R_b , and R_t is not still enough. R_a , R_b and R_t are resistors out of the device? What kind of resistors are used? what are their resistance values? How can you assume that all resistors are in the same ambient temperature?

Response #3

Thanks for your comment.

In Figure S1, R_a , R_b , and R_t are the fixed-value resistors mounted on the circuit board out of the sensor device, and they are the standard resistors with a very low temperature coefficient of resistance (TCR=5 ppm/°C). The fixed resistors are utilized to balance the Wheatstone bridge, and their resistance values are determined by the following equations:

$$\begin{cases} R_a = R_{h0} \\ R_b = R_{c0} \\ R_t = \Delta T \cdot \alpha \cdot R_{c0} \end{cases}$$

where R_{h0} and R_{c0} are the resistances of the hot-film and the cold-film at 0 °C respectively, ΔT is the temperature difference between the hot-film and the cold-film (whose temperature is the same as the ambient temperature), α is the temperature coefficient of resistance (TCR) of the hot-film and cold-film. The detailed derivation process can be seen on Page 2-3 in the Supplementary materials. Since R_a , R_b , and R_t are mounted in the same area on the circuit board, and thus in the same ambient temperature T . It is worth mentioning that R_a , R_b , and R_t are barely temperature-sensitive as they have very low temperature coefficients of resistance (TCR=5 ppm/°C). The corresponding content and revisions can be seen on Page 2-4 in the revised Supplementary Materials.

Comment #4

Basically, the effect of temperature variation is minimized if all four active strain gauges form the Wheatstone bridge circuit. Most strain-gauge load cells use this principle. In the suppl. material, authors claims that the temperature difference between hot and the environment is constant, but no evidence and explanation about the temperature independence about R_t is not provided. According to the last equation in the suppl. mater., ΔT is proportional to R_t . I also couldn't understand how the Wheatstone bridge circuit in Figure S1 could be balanced. R_h is about 50 Ω and R_c is about 500 Ω . R_t is connected to R_c , then what would be resistance values of R_a and R_b ?

Response #4

Thanks for your comment.

As we have mentioned above, since R_t is a fixed-value resistor installed on the circuit board independently of the sensor, and it is barely temperature-sensitive because it has a very low temperature coefficient of resistance (TCR=5 ppm/°C), much smaller than that of the hot-film R_h and cold-film R_c (TCR \approx 2200 ppm/°C). Therefore, $R_t = \Delta T \cdot \alpha \cdot R_{c0}$ ensures ΔT is constant.

For the Wheatstone bridge in Figure S1, it is a closed-loop feedback control circuit that ensures the Wheatstone bridge is self-balanced, achieving the following relationship:

$$\frac{R_a}{R_b} = \frac{R_h}{R_c + R_t}$$

where the resistances of R_a , R_b , and R_t are configured as follows:

$$\begin{cases} R_a = R_{h0} \\ R_b = R_{c0} \\ R_t = \Delta T \cdot \alpha \cdot R_{c0} \end{cases}$$

It is mentioned that the resistance of the cold-film (\sim 500 Ω) is much larger than the hot-film (\sim 50 Ω), thus the cold-film's Joule heat is much lower than the hot-film and can be ignored. Therefore, the cold-film temperature is nearly the same as the ambient temperature T .

Since the hot-film temperature of R_h is $T + \Delta T$ and the cold-film temperature of R_c is T , the bridge is balanced:

$$\frac{R_a}{R_b} = \frac{R_{h0}}{R_{c0}} = \frac{R_{h0}(1 + \alpha(T + \Delta T))}{R_{c0}(1 + \alpha(T + \Delta T))} = \frac{R_{h0}(1 + \alpha(T + \Delta T))}{R_{c0}(1 + \alpha T) + R_{c0} \cdot \alpha \cdot \Delta T} = \frac{R_h}{R_c + R_t}$$

The detailed explanation and the corresponding revisions can be seen on Page 2-4 in the revised Supplementary Materials.

Comment #5

Just scaling down the sensing area couldn't be a solution. In this case, the object could touch the other part of the sensing area. Actually, we can call the sensing device as "tactile sensor" if and only if the device has an array of sensing elements that could measure physical quantities upon touch over an area. If not, what is difference between "tactile sensor" and force sensor?

Response #5

Thanks for your comment.

Yes, an array design could be a solution to measure the physical quantities upon touch over an area. In this paper, we employ one sensing element in consideration of a small area of the robotic fingertip. We will consider utilizing an array of sensing elements for the tactile e-skin in our future work. In addition, we think the tactile sense should also be multimodal, like human skin. Therefore, we integrate multimodal tactile perceptions of pressure, temperature, matter thermal property, texture, and slippage into our tactile sensor.

Thanks again for your valuable comments and suggestions.

Reviewer #3

This manuscript details an interesting thermometry sensor that is used to perceive pressure, slipping, temperature, thermal conductivity, and surface texture for use with robotic systems. The sensor is further utilised in combination with visual data to present an approach for allowing a robot to undertake complex tasks.

Comment #1

I think that the revised manuscript is much improved, and my main issues, the lack of detail regarding aspects of the methodology and (critically) the lack of detail regarding the repeatability of measurements have been addressed to some extent. I did note that full company details were not provided in most cases (city and country of supplier). While I am happy to see the addition of error bars to Figure 2c and 2e, I am still not certain how many repeat measurements were taken to get these measurements (regarding temperature response, it just says that it was tested multiple time). Is the presented data representative for one specific sensor, with measurements repeated, or multiple sensors? How were the error bars calculated? This information needs to be present in the manuscript.

Response #1

Thanks for your comments and suggestions.

For the full company details, we have added the information of the cities and countries of suppliers in the Materials and Methods section.

For the error bars in Figures 2c and 2e, we have added corresponding descriptions in the text. Error bars shown in Figure 2c and 2e are the standard deviations of five repeated measurements of one specific tactile sensor.

The corresponding revisions can be seen on Page 6 and 16-17 in the revised Manuscript.

Comment #2

My other remaining minor criticism relates to the framing of the novelty of the work. I highlighted this in my previous review, and the authors have provided an excellent and detailed explanation of why the work is novel, however I still feel that in the manuscript this is somewhat unclear. I think that three or four extra lines in the introduction clearly expressing this would make the novelty of the manuscript clearer to the reader.

Response #2

Thanks for your comments and suggestions.

We have revised the introduction section and added the corresponding contents to highlight the novelty of the manuscript. As we have mentioned before, this paper mainly demonstrates new multimodal tactile sensing with superior sensing performance, a new tactile-visual fusion robot architecture, a new grasping strategy, a new tactile-visual fusion recognition strategy, and a new housekeeping application.

The corresponding contents and revisions can be seen on Page 3 in the revised Manuscript.

All of the above revisions have been marked in red font in the text of the revised Manuscript and revised Supplementary Materials.

Thanks for the comments again.

Sincerely

REVIEWERS' COMMENTS

Reviewer #3 (Remarks to the Author):

This manuscript details a novel thermometry sensor that is used to perceive pressure, slipping, temperature, thermal conductivity, and surface texture for use with robotic systems. The sensor is further utilised in combination with visual data to present an approach for allowing a robot to undertake complex tasks.

The authors have carefully considered and implemented my suggestions from the last round of reviews. I have no further suggested improvements.